# Antiviral Response across Genotypes after Treatment of Chronic Hepatitis B Patients with the Therapeutic Vaccine NASVAC or Pegylated Interferon

**DOI:** 10.3390/vaccines11050962

**Published:** 2023-05-09

**Authors:** Mamun Al-Mahtab, Sheikh Mohammad Fazle Akbar, Osamu Yoshida, Julio Cesar Aguilar, Gerardo Guillen, Yoichi Hiasa

**Affiliations:** 1Interventional Hepatology Division, Department of Hepatology, Bangabandhu Sheikh Mujib Medical University (BSMMU), Dhaka 1000, Bangladesh; 2Department of Gastroenterology and Metabology, Ehime University Graduate School of Medicine, Ehime 791-0295, Japan; 3Center for Genetic Engineering and Biotechnology, Havana 10400, Cuba

**Keywords:** chronic hepatitis B, NASVAC, HBV genotype D, pegylated interferon, immune therapy

## Abstract

An open-level, randomized and treatment-controlled clinical trial has shown that a therapeutic vaccine containing hepatitis B surface antigen (HBsAg) and hepatitis B core antigen (HBcAg) (NASVAC) is endowed with antiviral and liver protecting capacity and is safer than pegylated interferon (Peg-IFN) in patients with chronic hepatitis B (CHB). The present study provides information about the role of the hepatitis B virus (HBV) genotype in this phase III clinical trial. From a total of 160 patients enrolled in this trial, the HBV genotypes of 133 patients were characterized, and NASVAC induced a stronger antiviral effect (HBV DNA reduction below 250 copies per mL) than Peg-IFN. The antiviral effects and alanine aminotransferase levels were not significantly different among different HBV genotypes in NASVAC-treated patients. However, a significantly higher proportion of genotype-D patients receiving NASVAC showed better therapeutic effects, compared to genotype-D patients receiving Peg-IFN, with a marked difference of 44%. In conclusion, NASVAC seems to be a better alternative to Peg-IFN, especially in patients with HBV genotype-D patients. This reflects the attractiveness of NASVAC in countries where genotype D is highly prevalent. The mechanisms underlying the effect of HBV genotype are being studied in a new clinical trial.

## 1. Introduction

Out of a population of 7.5 billion, approximately one-third of all people worldwide has been infected by the HBV at some point of their life, and this can be confirmed using serological tests. The World Health Organization (WHO) estimates that approximately 2 billion have been infected by the HBV [1]. However, most of these people recover, either asymptomatically or after an acute attack of icterus jaundice [1,2,3]. Out of this large number of cases, an estimated 296 million people are chronically infected by HBV, as reported by the WHO. About 25% of these chronically infected subjects are prone to developing progressive liver diseases, such as cirrhosis of the liver and/or hepatocellular carcinoma. HBV-related pathologies accounted for an estimated 820,000 deaths in 2019. 

Peg-IFN and nucleos(t)ide analogues (NUCs) are the currently recommended treatments for CHB infection. Peg-IFN reduces viral replication by stimulating an innate immune response and has the advantage of higher sustained off-therapy response rates; however, it is expensive and can have considerable side effects. NUCs interfere with the viral polymerase and efficiently suppress HBV viremia. However, even after prolonged treatment with NUCs (>5 years), virological relapse is unlikely after therapy discontinuation, a process hampered by potential side effects in a proportion of patients that develop increased alanine aminotransferase (ALT), leading to hepatic decompensation in some cases. Therefore, a quasi-eternal therapy is often recommended [4,5,6]. However, in practice, this precaution is often not adhered to for economic reasons. Thus, irregular medication with NUCs is among the major contributors to acute-on-chronic liver failure, both in chronic and cirrhotic patients [7].

In HBV infection, strong and multi-specific antiviral immunity is related to viral control. Consequently, the tolerance subversion and the modulation of the host immunity are the goals of immunotherapies. Both Peg-IFN treatment and therapeutic vaccination approaches have been recommended as new and novel therapeutic options [8].

The therapeutic vaccine NASVAC was designed as a formulation for nasal and subcutaneous administration and produced in GMP-certified areas. This vaccine formulation comprises recombinant surface (HBsAg) and nucleocapsid (HBcAg) antigens from an HBV genotype-A source. This novel product was pharmacologically tested in different animal models and completed all CMC, preclinical stability, toxicology and clinical development steps up to phase III [9,10]. The first sanitary registrations were granted in countries where the product was clinically developed, and some others implemented clinical trials and sanitary registration steps. The superiority of NASVAC in terms of safety and the reduction in HBV DNA below 250 copies per mL was demonstrated in a phase III trial [10]. However, the effect of the therapeutic vaccination across different viral genotypes remains unknown.

HBV genotype distribution in Bangladesh mostly exhibited a prevalence of HBV genotypes C and D in some small observational studies. A well-designed study by Raihan et al. explored these HBV genotypes in 360 patients with HBV infection in Bangladesh. The study found that three HBV genotypes are prevalent in Bangladesh: HBV genotype A (18%); HBV genotype C (43%); and HBV genotype D (39%) [11]. 

In the present report, patients enrolled in the phase III clinical trial were characterized in terms of viral genotype. Antiviral responses was studied across genotypes for NASVAC-vaccinated and Peg-IFN-treated patients. In addition, the behavior of the biochemical variable ALT was also analyzed across genotypes in vaccinated patients, considering the characteristic increasing pattern of homogeneous ALT found in vaccinated patients, as well as the direct relation between the induction of specific anti-HBV immunity and the increase in ALT values. This study provides insights regarding the potential use of NASVAC in some special populations of CHB patients.

## 2. Materials and Methods

### 2.1. Phase III Clinical Trial

All patients enrolled in this study were diagnosed with CHB based on historical and clinical data, as well as serological, biochemical, virological, and imaging assessments. They were treatment-naïve and had not received any antiviral or immune stimulatory medication for their illness. The levels of alanine aminotransferase (ALT) had to be above the upper limits of normal (ULN, >42 U/L). The levels of HBV DNA were higher than 10^3^ copies/mL for hepatitis B e antigen (HBeAg(−)) patients and 10^4^ for HBeAg(+) patients. Patients were excluded from the study if they were in an immune-tolerant or inactive carrier state with normal ALT. Additionally, patients with advanced liver disease with cirrhosis and/or HCC were excluded. Patients with positive serology for hepatitis C, hepatitis delta or human immune-deficiency virus were excluded. CHB patients who were critically ill, patents with hypertension, hyperthyroidism, epilepsy, malignancies or any non-controlled systemic disease were also excluded. Additionally, pregnant patients or nursing women were excluded from the trial. CHB patients with very high levels of ALT (ALT above 500 U/L) were also excluded. 

### 2.2. Therapeutic Vaccine NASVAC

NASVAC is the first therapeutic vaccine licensed for the treatment of CHB. In 2016, after receiving the permission from Cuban regulatory authorities, it was introduced in the Cuban health system. Manufactured under GMP conditions, this product is a 1:1 formulation of 100μg of HBsAg (Pichia pastoris-derived recombinant HBsAg subtype adw2) and 100μg of HBcAg (purified Escherichia coli-expressed recombinant full-length HBcAg), both from a genotype-A-infected patient. NASVAC is administered by intranasal (IN) and subcutaneous (SC) routes. This product was designed, produced, and developed by the CIGB (Havana, Cuba) and has been tested as a monotherapy in Cuba, Bangladesh and Japan.

The overall objective of this study was to assess the safety and efficacy of NASVAC in CHB patients. The mode of therapy (via the nasal route) was chosen due to several factors such as targeting the mucosal immunity-based vaccine production. In addition, the nasal route was selected as this can be easily used in developing and resource-constrained countries; only injection-based vaccines cannot be used by the majority of population of those countries. However, these countries harbor the main bulk of CHB patients. Thus, treating these patients in a non-parental mode is an important variable for treating CHB patients and achieving “Elimination of Hepatitis by 2030”, a goal of WHO. The injection route was chosen to activate the systemic compartment of the patients by specific antigens. In fact, the safety and efficacy of both nasal and injection routes were optimized in a phase I/II clinical trial in which the mechanism of action was also studied in peripheral blood lymphocytes and antigen-presenting dendritic cells. 

NASVAC was developed and optimized for repeated administration; it does not include adjuvants or preservatives. The immunogenicity of NASVAC depends on the intrinsic characteristics of HBsAg and HBcAg virus-like particles (VLP) as well as on the interaction of both antigens in phosphate-buffered saline (PBS). The final product undergoes extensive quality control tests before release, including assays for sterility, endotoxin contents, identity and purity, in order to ensure that the product is safely formulated and effective [9,10]. A phase I/II clinical trial with NASVAC was conducted in Bangladesh to ascertain the safety and efficacy of NASVAC in CHB patients [9].

### 2.3. Design of the Phase III Clinical Trial

Subsequently, a phase III, treatment-controlled, open-label and randomized clinical trial was conducted by a group of scientists from Bangladesh at Bangabandhu Sheikh Mujib Medical University (BSMMU) (Dhaka, Bangladesh) and Farabi Hospital (Dhaka, Bangladesh) in compliance with the Declaration of Helsinki and with the principles of good clinical practice, certified by regulatory authorities after in situ audits. The study was registered in ClinicalTrials.gov (NCT01374308) and was published recently [10]. 

Briefly, all patients enrolled in the clinical trial were diagnosed with CHB on the basis of serological, biochemical, virological, and imaging assessments. All patients were treatment naïve and none received any immune stimulatory medication. The age of the patients ranged from 18 to 65 years, and patients of both sexes were enrolled. All of them expressed HBV DNA in sera and HBsAg in sera for more than 6 months. The levels of ALT were above the upper limits of normal (ULN) values during or before enrolment. HBV levels were fluctuating but were higher than 10^3^ copies/mL for hepatitis B e antigen [HBeAg(−)] patients and 10^4^ for HBeAg-positive [HBeAg(+)] patients. Patients were excluded from the study in case of decompensated liver disease, asymptomatic HBV carriers, cirrhosis or detection of hepatocellular carcinoma; positive serology for HCV, HDV or HIV, among a number of inclusion and exclusion criteria [9,10].

Although HBV genotypes A, C and D are prevalent in Bangladesh, no patients were coinfected with more than one genotype, particularly genotypes A, C, and D. 

Out of the 360 patients with CHB that appeared for pre-recruitment, 160 CHB patients were finally selected for the study after meeting inclusion and exclusion criteria. The patients were randomly divided into two groups (1:1). Eighty patients were assigned to receive 180 μg of Peg-IFN (PegHeberon, Peg-IFN alpha 2b, Center for Genetic Engineering and Biotechnology, (CIGB), Havana, Cuba) once weekly for 48 consecutive weeks. The other 80 patients received NASVAC. Therapeutic vaccination was conducted in two cycles. In the first cycle, 1.0 ml of NASVAC was administered via the intranasal route using a nasal spray on five occasions at 2-week intervals. In the second cycle, 1 month apart, the same vaccine formulation was simultaneously administered by IN and SC routes (the same dose was administered by both routes: 1.0 ml containing 100 μg of HBsAg and 100 μg of HBcAg. All patients were observed for 2 h after each vaccination and periodically after vaccination. Serum was collected from each patient before the study commenced, before each vaccination, after five nasal vaccinations (after the end of first cycle) and after the end of second cycle (end of treatment—EOT). Follow-up studies were conducted 24 weeks after the end of each treatment. 

### 2.4. Safety and Efficacy Evaluation 

Adverse reactions were immediately measured and up to 2 h after immunization. Any adverse event during the inter-immunization periods was also recorded before the administration of the next dose of the products. Adverse events such as fever, weakness, general malaise, headache, local pain, nausea, loose motion, hair loss, gastrointestinal disorder, fatigue, anxiety, dyspepsia, gum bleeding, aphthous ulcer, skin rash, vomiting, and bitter taste were checked in all patients. In addition, an ear, nose and throat (ENT) specialist checked each patient for abnormal pathogenicity in the nose and similar areas.

In addition to monitoring adverse events, blood from all patients was tested to assess hematology and the general parameters of the inflammatory responses, kidney function, and liver function. Sera were also collected at EOT, and 24 weeks after the EOT to assess the long-term safety of the studied products. 

Efficacy was measured quantifying the levels of HBV DNA and ALT, as well as the qualitative detection of HBeAg at different points during and after treatment. All assessments were made at tertiary-level reference laboratories in Dhaka, Bangladesh using a standard methodology with good clinical practices, as assessed via monitoring and regulatory audits. 

### 2.5. Genotype Characterization

A total of 149 samples were assayed at Hôtel Dieu, Service d’hépatologie et de gastroentérologie, in Lyon, France. HBV genotype was assessed using the INNOLiPA Assay (Innogenetics, Gent, Belgium) as part of an international collaboration with the laboratory of Dr. Christian Trepo.

### 2.6. Statistical Analysis

The methodology for analyzing the primary (quantitative serum HBV) and secondary response variables (biochemical markers, hematology, and clinical biochemistry) has previously been reported [10]. For the analysis of quantitative control variables (age, weight, body mass index (BMI), duration of the disease, initial viral load), the central trend statistical and dispersion values were estimated for the study groups. A normality analysis and quality-of-fitness estimation were also carried out, and IC 95% was estimated using Bayesian methods for the difference between groups. On the other hand, contingency tables were designed that correspond to each qualitative control variable (sex, smoking, and other vices), showing the respective frequencies and percentages of treatment groups. The software used for the statistical analysis were SPSS (version 15.0 for Windows), EPIDAT 3.1 and NCSS-PASS-GESS. 

The results of studying the relationship of the viral genotype with the main variable of response to treatment (quantitative HBV DNA) and with ALT are shown in a contingency table with the respective frequencies and percentages between the variables according to treatment group. The χ-square was estimated to evaluate the dependence relationship between these variables in each treatment group. The PRISM 5 software was used for statistical analysis.

## 3. Results

The baseline characteristics of the patients enrolled in the clinical trial are presented in Table 1. The characterization of genotypes was conducted in 149 patients with available samples from the 158 patients that started the study. Sixteen patients were tested, but their genotype was indeterminable due to technical reasons (five in NASVAC and eleven in Peg-IFN groups). Ultimately, the genotypes of 133 patients were successfully characterized. 

In general, the genotypes with a higher proportion of patients were genotypes C and D, prevalent in Bangladesh. There was no significant difference between treatment groups in the proportion of patients represented by each genotype; however, there is an increased proportion of patients with genotype A and a lower proportion of genotype D in the vaccine group (Table 1).

No severe adverse event was recorded in patients that ceased treatment schedules. However, the incidences of fever, weakness, general malaise, headache, and local pain were significantly higher in patients who received Peg-IFN compared to those who received NASVAC [10]. With respect to the antiviral effect across genotypes per study group, it a higher proportion of patients independently treated with NASVAC was found to have less than 250 copies/mL in the case of viral genotypes 24 weeks after the end of each treatment. There was no difference in the frequency of responding patients per genotype in patients vaccinated with NASVAC. However, an impact of HBV genotype was recorded when comparing the antiviral effect of CHB patients with HBV genotype D between NASVAC recipients and Peg-IFN recipients. The antiviral effect of NASVAC was relatively higher in patients with genotype D among vaccinated patients and, in contrast, genotype-D patients had the lowest response in the group of patients treated with Peg-IFN (Figure 1), reaching a remarkable 44% net difference and statistical differences between treatments (*p* < 0.05). 

The study of the biochemical response revealed a homogeneous ALT flare (2 to 5 times ULN; 42 U/L is the ULN) in the majority of patients receiving therapeutic vaccination (NASVAC) as compared to the patients receiving Peg-IFN. Only few Peg-IFN-treated patients experienced such increases in ALT. The analysis of the ALT variable across genotypes in NASVAC recipients revealed a similar ALT increase among different genotypes (Figure 2). The means of the ALT peak values for all genotype groups among vaccinated patients, independent of time, revealed a very similar level of intensity. The effect of the reduction in ALT intensity at week 12 for genotype-D-vaccinated patients was compensated by previous peaks of ALT that appeared at week 8. In summary, a faster ALT increase was detected for genotype-D-vaccinated patients compared to patients vaccinated against the other two genotypes (Figure 2).

Patients with chronic hepatitis B with HBV genotype D exhibited better antiviral responses than those of HBV genotype CHB patients treated using Peg-IFN. 

The therapeutic vaccine candidate induced a stronger antiviral suppression.

## 4. Discussion

A remarkable difference was found between treatments in genotype-D patients, and such findings are relevant for a number of reasons. First, PegIFN is the only finite treatment registered worldwide and has the highest off-therapy effect; however, it is not widely used due to safety concerns. Second, this is the first time that a therapeutic vaccination has achieved promising results in a phase III clinical trial, and these antiviral results were balanced among genotypes A, C and D. Phase III results are consistent with data on safety and antiviral potentials, as well as the control of liver damages in other clinical trials reported for phase I/II clinical trial with NASVAC [9]. Third, the pattern of antiviral response across genotypes induced by Peg-IFN is also consistent with the pattern of response across genotypes reported by a previous meta-analysis, concluding that the interferon treatment of genotype-A patients has a better response compared to genotype-D patients, regardless of HBeAg status.

A significant difference in the proportion of patients with less than 250 copies/mL at 24-week follow-up was found in genotype-D patients in the two treatments. This was caused by the higher response found in NASVAC-treated patients, combined with lower responses in Peg-IFN-treated genotype-D patients (Figure 1). Although the sample size was not that large (22 in NASVAC and 29 in Peg-IFN), the difference was remarkable (44%) leading to the detection of significant differences in favor of NASVAC. Although a larger number of patients are required to further validate such results, it was encouraging to find a relatively balanced number of patients in the three genotypes detected in Bangladesh, which is not a common epidemiological situation worldwide [11,12,13,14,15].

Therapeutic vaccination is an attractive approach for the first-line therapy due to its finite character and remarkable safety. Although genotype D is considered a pandemic genotype, its prevalence is very high in large countries and regions. This is the case for Russia (93% prevalence) [16]; Northern Africa; Central, Western and Southern Asia; and Southern and Eastern Europe, where sometimes genotype D accounts for almost all patients, as recently described in a review article. In addition, a faster disease progression to liver cirrhosis and HCC can be associated with infections with genotypes C, D and F [11], further stressing the importance of these results for studying the introduction of NASVAC as a first-line treatment in patients from these regions. In summary, all genotypes showed superiority when therapeutic vaccination was compared to Peg-IFN treatment; however, some countries may benefit even more from this remarkable difference between NASVAC and Peg-IFN due to their very high prevalence of HBV genotype D. 

One additional point of interest was found in the analysis of ALT in NASVAC-treated patients. By week 12, a peak of ALT was reported in almost all vaccinated patients [9]. The HBV is a non-cytopathic virus, which means that it is unable to lysate cells. Thus, an increase in ALT reflects the degree of immune activation in response to viral detection. We have shown that NASVAC is able to contain the HBV replication and liver damage in a non-cytopathic manner by conducting serial follow-up [17,18,19]. 

A higher proportion of genotype-D patients reacted to vaccination with ALT peaks at week 8 instead of exhibiting normal behavior at week 12. The significant increase in ALT mean value at week 8 compared to week 0 for genotype-D patients may be related to the higher antiviral effect found 24 weeks after the end of follow-up. On the other hand, the net ALT mean was reduced at week 12 for genotype-D patients; however, the mean of the ALT peaks did not differ among genotypes, suggesting a similar intensity of immune stimulation among patients from genotypes when considering the surrogate value of ALT in relation to immune response. In fact, no significant differences in antiviral effect were found between different genotypes of NASVAC-treated patients. 

A long-term follow-up analysis of patients is required to further determine the long-term clinical effect of the described results, exploring the predictive value of variables such as viral genotype or ALT. Larger clinical trials in the region may also increase the sample size in patients with similar genetic backgrounds and characteristics. However, the study of patients with different ages, co-morbidities, and genetic differences will complement this analysis. 

In conclusion, treatment with NASVAC is similarly effective in CHB patients of three of the major genotypes. Additionally, the levels of alanine aminotransferase kinetics are similar among all three genotypes. The remarkable difference in Peg-IFN treatment specifically for antiviral effects on genotype-D patients generates an attractive scenario for countries where genotype D has a higher prevalence. 

## Figures and Tables

**Figure 1 vaccines-11-00962-f001:**
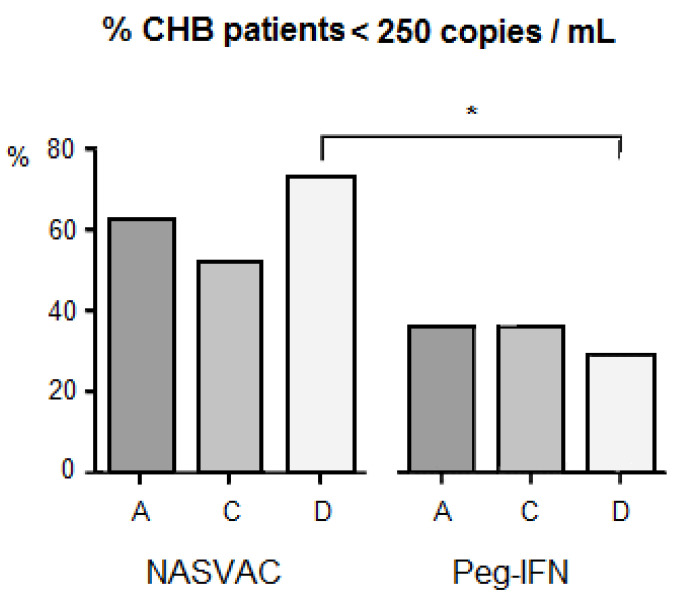
Impact of HBV genotype diversity on HBV DNA negativity under various treatment modalities in patients with chronic hepatitis B. The letters “A”, “C” and “D” at the lower part of horizontal line indicate HBV genotype A, HBV genotype, C and HBV genotype D, respectively. *: *p* < 0.05.

**Figure 2 vaccines-11-00962-f002:**
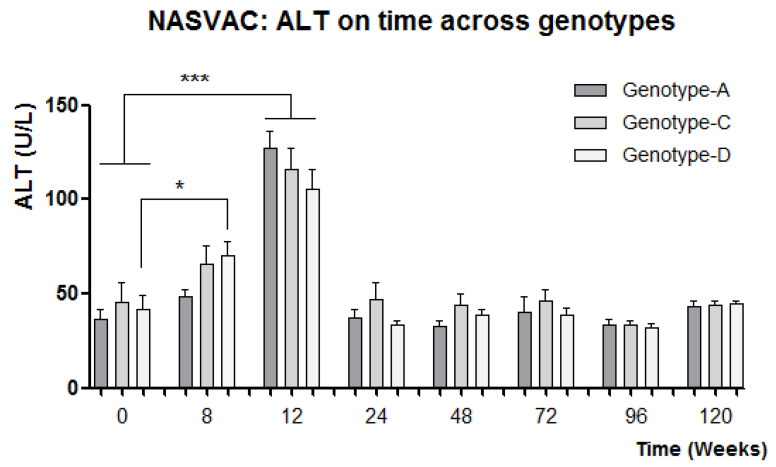
Relationship between ALT kinetics in different HBV genotypes. HBV genotype-A, HBV genotype-B, HBV-genotype C has been shown by bars. * *p* < 0.05: statistically significant differences; ***: *p* < 0.001, very significant differences (One-way ANOVA followed by Tukey’s multiple comparison test).

**Table 1 vaccines-11-00962-t001:** The baseline information of patients.

Variables		NASVAC		Peg-IFN		Stats
		N	%	N	%	*/ns
Total	After 1st dose	78	49.4	80	50.6	ns
Gender	Female	14	17.9	10	12.5	ns
	Male	64	82.1	70	87.5	ns
Age (years)	Mean ± DS Median ± RQ (Min; Max)	29 ± 7 28 ± 10 (18; 50)		29 ± 7 28 ± 11 (18; 48)		ns
Height (m)	Mean ± DS Median ± RQ (Min; Max)	1.51 ± 0.10 1.52 ± 0.00 (1.22; 1.83)		1.52 ± 0.06 1.52 ± 0.00 (1.22; 1.83)		ns
Weight (Kg)	Mean ± DS Median ± RQ (Min; Max)	59 ± 10 59 ± 16 (36; 81)		61 ± 10 61 ± 15 (35; 85)		ns
Body mass index (Kg/m^2^)	Mean ± DS Median ± RQ (Min; Max)	26.01 ± 4.43 25.78 ± 6.83 (15.50; 36.85)		26.27 ± 4.51 26.36 ± 6.43 (17.22; 40.26)		ns
HBV DNA (log(copies/mL)) at baseline	Mean ± DS Median ± RQ (Min; Max)	5.4 ± 2.1 4.7 ± 1.8 (3.2; 13.0)		5.8 ± 2.3 5.2 ± 2.6 (3.1; 12.5)		ns
ALT (U/L), at time zero	Mean ± DS Median ± RQ (Min; Max)	42.3 ± 42.6 30.0 ± 22.0 (10.0; 262.0)		44.7 ± 32.2 37.0 ± 19.8 (10.0; 226.0)		ns
HBeAg(+) at baseline	No. (%)	15 (19.2)		18 (22.5)		ns
Genotyped patients	No. (%)	69 (52)		64 (48)		ns
Genotype-A	No. (%)	20 (29)		11 (17)		ns
Genotype-C	No. (%)	27 (39)		24 (38)		ns
Genotype-D	No. (%)	22 (32)		29 (45)		ns

ns: non-significant statistical differences; *: significant differences (*p* < 0.05).

## Data Availability

The raw data supporting the conclusions of this article will be made available by the authors, without undue reservation.

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
