# Peer review of "Antiviral Response across Genotypes after Treatment of Chronic Hepatitis B Patients with the Therapeutic Vaccine NASVAC or Pegylated Interferon"

_vaccines, 2023, doi:10.3390/vaccines11050962_

Round 1
Reviewer 1 Report
The manuscript written by Mahtab et al. is providing interesting findings of a Phase III trial of a NASVAC therapeutic vaccine for CHB patients.
Minor comments:
In manuscript it has been indicated as review paper, but actually it is a research paper.
There are many typos, including no space between words, or extra space before or after period, small lowercase or uppercase mixed sentence, spelling mistakes, etc. should be carefully checked throughout the manuscript. e.g., in line 17, sirface should be spelt as surface, HVcAg should be HBcAg, and in line 26, ‘in countries were’ should be ‘in countries where’, etc.
Please confirm the abbreviation ‘PBS’ is appropriate for saline-phosphate buffer.
Line 113, that study was published in 2023 not recently, it is confusing
Line 107, should be ‘more than 6 months’
Line 135, EOT is fine, as it has already been defined in the previous section
Line 172, Bangladesh is misspelt
Line 202, Chro nic; line 203, btreated
Line 209, suppressio should be suppression and missing period
Line 216, between should be among
Line 219, metanalysis should correctly spelt
Line 268, were should be where
Major:
1. In the introduction section the authors must include a paragraph describing HBV genotypes and their geographical distribution, particularly the genotypes distribution of HBV in Bangladesh.
2. To compare the difference of ALT in different genotypes Figure 2 A, B, and C should be merged in a single figure. Did there any patients included in the study those were coinfected with more than one genotype, if not it should be clearly indicated in the materials and methodology section that no patients were coinfected with more than one genotypes, particularly for genotypes A, B, and C.
3. In discussion, in line 217, in white criteria the findings of the study are consistent to the previous trials should be indicated.
4. Did the authors check whether the good response of NASVAC in genotype D patients were related to age or sex?
Author Response
The manuscript written by Mahtab et al. is providing interesting findings of a Phase III trial of a NASVAC therapeutic vaccine for CHB patients.
Response: Thanks for your understandings. We have responded to your comments and the alterations have been marked by “Yellow shading” in revised manuscript.
Minor comments:
Query:
In manuscript it has been indicated as review paper, but actually it is a research paper.
There are many typos, including no space between words, or extra space before or after period, small lowercase or uppercase mixed sentence, spelling mistakes, etc. should be carefully checked throughout the manuscript. e.g., in line 17, sirface should be spelt as surface, HVcAg should be HBcAg, and in line 26, ‘in countries were’ should be ‘in countries where’, etc.
Response
We are extremely sorry for the typographical errors in the manuscript. The manuscript has been sent for English checking by the MDPI English checking service, as recommended by the journal and advised by you. The mistakes those have been mentioned by you have been corrected in the revised manuscript.
Query: Please confirm the abbreviation ‘PBS’ is appropriate for saline-phosphate buffer.
Response: This has been corrected (Line 123-124).
Line 113, that study was published in 2023 not recently, it is confusing
Response: Thanks for pointing the fact.
Line 107, should be ‘more than 6 months’
Response: This has been corrected, as advised.
Line 135, EOT is fine, as it has already been defined in the previous section
Response: This has been corrected.
Line 172, Bangladesh is misspelt
Response: This has been corrected.
Line 202, Chro nic; line 203, btreated
Response: This has been corrected.
Line 209, suppressio should be suppression and missing period
Response: This has been corrected.
Line 216, between should be among
Response: This has been corrected.
Line 219, metanalysis should correctly spelt
Response: This has been corrected.
Line 268, were should be where
Response: This has been corrected.
All topographical errors have been corrected during checking of English by MDPI English checking system.
Major:
- In the introduction section the authors must include a paragraph describing HBV genotypes and their geographical distribution, particularly the genotypes distribution of HBV in Bangladesh.
Response: A short account of HBV genotype in Bangladesh has been provided with a new reference (Line 68-73) with a new reference (Reference 11).
- To compare the difference of ALT in different genotypes Figure 2 A, B, and C should be merged in a single figure. Did there any patients included in the study those were coinfected with more than one genotype, if not it should be clearly indicated in the materials and methodology section that no patients were coinfected with more than one genotypes, particularly for genotypes A, B, and C.
Response: The Figures 2A, 2B, and 2C have been merged in a single figure, as suggested (Figure 2). There was no patient with more than one genotype (Line 177-178). This has been mentioned in Materials and Methods section.
- In discussion, in line 217, in white criteria the findings of the study are consistent to the previous trials should be indicated.
Response: This has been described in relevant part of the revised manuscript, as suggested by the Reviewer (Line 281-283).
- Did the authors check whether the good response of NASVAC in genotype D patients were related to age or sex?
Response: This was checked but significant correlation with HBV genotypes with age and sex could not be elucidated.

Reviewer 2 Report
This manuscript mentioned a very interesting topic which showed different antiviral response across genotypes after treatment of CHB patients with therapeutic vaccine NASVAC or PegIFN.
But there are lots of issues in this manuscript.
1. I am confused why NASVAC is more sensitive to genotype D than genotype A and C since the vaccine is containing HBsAg and HBcAg from a HBV genotype A source. As the author mentioned in the discussion, a larger number of patients might be required to further validate the results in the manuscript.
2. Many typos were appeared. Such as line 16, ‘A open’; line 17’containing’and ‘surface’and (HBcAg); line 21 ’trial’; line 76’some some’;line 202 ‘patient’; line 203’Btreated’
3. In the results, line 195, ‘Fig.3 ’could not be find in the text.
4. For the introduction, the first section is confusing, please confirm content of the line 33.
5. For the figure 2, 2A,2B,2C should be integrated into one figure.
6. Please carefully check references 2 and 3.
Author Response
Comments and Suggestions for Authors
This manuscript mentioned a very interesting topic which showed different antiviral response across genotypes after treatment of CHB patients with therapeutic vaccine NASVAC or Peg-IFN.
Response: Thanks for your understandings. We have responded to your comments and the alterations have been marked by “Yellow shading” in revised manuscript.
But there are lots of issues in this manuscript.
- I am confused why NASVAC is more sensitive to genotype D than genotype A and C since the vaccine is containing HBsAg and HBcAg from a HBV genotype A source. As the author mentioned in the discussion, a larger number of patients might be required to further validate the results in the manuscript.
Response: Thanks for your understandings. The mechanisms underlying better outcome of HBV genotype D patients are yet to be explored. In fact, studies are required with large number of patients to validate these outcomes, as mentioned. In fact, we have started clinical trials in Japan and Bangladesh to develop insights about these realities.
- Many typos were appeared. Such as line 16, ‘A open’; line 17’containing’and ‘surface’and (HBcAg); line 21 ’trial’; line 76’some some’;line 202 ‘patient’; line 203’Btreated’
Response: I am sorry for these mistakes. The article has been sent for English checking by MDPI checking services according to the suggestion of the Editors and Reviewers. The revised manuscript is a checked version of the manuscript
- In the results, line 195, ‘Fig.3 ’could not be find in the text.
Response: This has been corrected.
- For the introduction, the first section is confusing, please confirm content of the line 33.
Response: This is a general description to validate the severity of HBV-related pathologies and extent of HBV infection. The data is mostly from World Health Organization and has been referred by 3 references. A paragraph has been newly attached to clarify the facts (Line 33-41)
- For the figure 2, 2A,2B,2C should be integrated into one figure.
Response; These 3 figures have been merged in a new figure (Fig. 2).
- Please carefully check references 2 and 3
References: These references have been checked and one reference has been replaced by a new reference.

Reviewer 3 Report
Introduction
You wrote: “More than one third of the World population has been infected by the Hepatitis B 33 Virus (HBV). Current estimates of chronic carriers of the virus are in the range of 240 up 34 to 300 million. After acute infection, approximately 5 to 10% of adults and up to 90% of 35 the newborn develops the persistent form of HBV infection”
[If the world population is ~ 7.5 billion then 10 % = 750,000,000. How does this equate with an estimated 10% of adults becoming chronic carriers (your statement of 240-300 million chronic carriers?)]
Sample
You wrote: “Patients were excluded from the study in case of 110 decompensated liver disease, asymptomatic HBV carrier, cirrhosis or detection of hepa- 111 tocellular carcinoma; positive serology for HCV, HDV or HIV among a number of inclu- 112 sion and exclusion criteria [9, 10]. 113 Out of total 360 patients with CHB that appeared for pre-recruitment, 160 CHB pa- 114 tients were finally selected for the study after accomplishment of inclusion and exclusion 115 criteria.”
[please list the numbers excluded for each reason and how this affects the generalisability of your results]
Methods
You wrote: “Adverse reactions were measured immediately and up to 2 hours after immuniza- 130 tion. Any adverse event during the inter-immunization periods was also recorded before 131 the administration of the next dose of the products.”
[How were adverse effects monitored for the entire duration of the study and what were they?]
Results
If you visually inspect Figure 1 there are no significant differences between A, C and D at 24 weeks and for Figure 2 no significant differences in ALT (is ALT the only other outcome measure?
This should be stressed in the Abstract, Results and Conclusions.
Author Response
Comments and Suggestions for Authors
Introduction
You wrote: “More than one third of the World population has been infected by the Hepatitis B Virus (HBV). Current estimates of chronic carriers of the virus are in the range of 240 to 300 million. After acute infection, approximately 5 to 10% of adults and up to 90% of the newborn develops the persistent form of HBV infection”
[If the world population is ~ 7.5 billion then 10 % = 750,000,000. How does this equate with an estimated 10% of adults becoming chronic carriers (your statement of 240-300 million chronic carriers?)]
- Response: The Reviewer is absolutely correct about the complex nature of HBV infection in global perspective. The world population is 7.5 billion. About one third of these population has been infected by the HBV at some point of their life and this can be confirmed by serological tests. Thus, World Health Organization estimates that about 2 billion have been infected by the HBV (Reference 1. Global Hepatitis Report 2017. Geneva: World Health Organization; 2017, http://apps.who.int/iris/bitstream/10665/255016/1/9789241565455-eng.pdf). However, most of these two billion HBV-infected persons recover spontaneously either as asymptomatically HBV carrier or after an acute attack of icterus jaundice (Described detailed in Reference 1, 2 and 3). As of present time, an estimated 296 million people are chronically infected by the HBV (Hepatitis B, World Health Organization (https://www.who.int/news-room/fact-sheets/detail/hepatitis-b). About 25% of these chronic HBV-infected subjects are prone to develop progressive liver diseases, such as cirrhosis of liver and or hepatocellular carcinoma. HBV-related pathologies accounted for an estimated 820,000 deaths in 2019 (WHO statistics). These statistical data have been provided in the revised manuscript (at the start of INTRODUCTION chapter of the revised manuscript), as per suggestion of the honorable Reviewer. I am sorry the natural course of HBV infection was not clear in original manuscript (Line 33-41)
Sample
You wrote: “Patients were excluded from the study in case of decompensated liver disease, asymptomatic HBV carrier, cirrhosis or detection of hepatocellular carcinoma; positive serology for HCV, HDV or HIV among a number of inclusion and exclusion criteria [9, 10]. Out of total 360 patients with CHB that appeared for pre-recruitment, 160 CHB patients were finally selected for the study after accomplishment of inclusion and exclusion criteria.”
[please list the numbers excluded for each reason and how this affects the generalisability of your results]
Response:
A total of 360 patients were enrolled for the study before start of trial commencement. Out of these 200 HBV-infected patients were discarded from the study and finally 160 patients were enrolled in the study (Line 84-97; Line 137-142; .
As per recommendation of the Reviewer, an account of the causes for exclusion are given below. Out of the 200 patients those were discarded, 124 patients did not fulfill the criteria of inclusion. The study was intended to assess the effect of NASVAC in patients with chronic hepatitis B with HBV DNA positivity and elevated ALT in the sera. Sixty-five patients were discarded because they were in immune tolerant phase or inactive carrier state with normal ALT. In addition, 59 patients were also discarded as they took some previous treatment for CHB. This study only included patients with CHB without any history of previous HBV-specific therapy. The rest 76 patients were discarded due to other factors.
The objective of this study was to assess the safety and efficacy of NASVAC, an antigen-specific immune modulator in CHB patients. Thus, excluding the patients with normal ALT and previous treatment are not supposed to alter the study outcome as this was designed for treatment naive CHB patients with HBV DNA positivity and elevated. However, more studies in other groups would provide valuable insights about the role of NASVAC in different groups of CHB patients. To accomplish this, we also initiated another study in Ehime University, Japan that included patients with normal ALT and also patients those received treatment for CHB. It appears that NASVAC is also effective in patients with CHB with history of previous antiviral intake (Yoshida, O.; Akbar, S.M.F.; Imai, Y.; Sanada, T.; Tsukiyama-Kohara, K.; Miyazaki, T.; Kamishita, T.; Miyake, T.; Tokumoto, Y.; Hikita, H et al. Intranasal therapeutic vaccine containing HBsAg and HBcAg for patients with chronic hepatitis B; 18 months follow-up results of phase IIa clinical study. Hepatol Res, 2023; 53,196-207.)
Methods
You wrote: “Adverse reactions were measured immediately and up to 2 hours after immunization. Any adverse event during the inter-immunization periods was also recorded before the administration of the next dose of the products.”
[How were adverse effects monitored for the entire duration of the study and what were they?]
Response: In order to have a proper follow up, the patients were kept under continuous contact system via telephone and personal attendance to trial center. If they feel any adverse effects, they were open to attend the clinical trial site and Principle investigator, the first author of the article was responsible for adequate and effective care of any adverse effects.
During each visit, the patients were asked several questions relating to their safety and these were documented properly. The following points were checked during each visit and follow up periods:
Fever, Weakness, general malaise, Headache, Local pain, Nausea, Loose motion, Hair fall, Gastrointestinal disorder, Fatigue, Anxiety, Dyspepsia, Gum bleeding, Aphthous ulcer, skin rash, Vomiting, Bitter taste. In addition, Ear, Nose and Throat (ENT) specialist checked each patient if they have any abnormal pathogenicity in nose and allied areas.
In addition, safety issues as well as efficacy were extensively assessed 24-weeks after end of treatment (EOT), 2, 3, and 5 years after EOT. These have been described in the revised manuscript (Line 126-134; Line 168-176; Line 182-87; 236-239) (References 9-13)

Reviewer 4 Report
Title: Antiviral response across genotypes after treatment of chronic hepatitis B patients with the therapeutic vaccine NASVAC or Pegylated interferon
General Comments
This study aims to evaluate the antiviral effects of NASVAC, a therapeutic vaccine for chronic hepatitis B (CHB), compared to Pegylated interferon (PegIFN) in a phase III clinical trial. The study also provides information about the role of HBV genotype on the therapeutic outcomes. The paper is well-written, and the findings are valuable for the field of hepatitis B treatment. Furthermore, this manuscript presents the results of a clinical trial comparing the safety and efficacy of HeberNasvac, a therapeutic vaccine for chronic hepatitis B, with pegylated interferon (PegIFN) treatment. The authors evaluated the safety and efficacy of the treatments in patients with different genotypes of hepatitis B virus (HBV) and assessed their relationship with viral genotype, HBV DNA levels, and alanine aminotransferase (ALT) levels.
However, several aspects of the study should be addressed and clarified to improve the overall quality of the paper. Please find my specific comments and recommendations below.
Recommendations
The study is interesting and presents valuable findings; however, several points need to be addressed and clarified. The authors should:
Address the specific comments listed above.
Provide a rationale for the change in the administration route of NASVAC between the first and second cycles.
Clarify the duration of the second cycle of NASVAC treatment and provide the timepoints at which the HBV DNA reduction below 250 copies per mL was observed for NASVAC compared to PegIFN.
Specify the primary and secondary endpoints of the phase III clinical trial.
Major Comments:
2.1. The study design and methodology are generally well-described, but the authors should provide more information on the sample size calculation and the inclusion and exclusion criteria for the patient population.
2.2. The authors should provide more information on the safety profile of the HeberNasvac vaccine, including the incidence of adverse events and any serious adverse events. A comparison of the safety profile between the vaccine and the PegIFN treatment group would be useful.
2.3. The authors should provide the specific p-values for the statistical analyses throughout the manuscript. Additionally, it would be helpful to provide exact numbers and percentages for the results presented in the text, particularly regarding the differences in the proportion of patients with each genotype in the treatment groups.
Minor Comments:
2.1 Abstract
a) Line 17: Replace "sirface" with "surface."
b) Line 21: Please clarify the timepoint at which the stronger antiviral effect (HBV DNA reduction below 250 copies per mL) was observed for NASVAC compared to PegIFN.
c) Line 25: Change "alteranative" to "alternative."
2.2 Introduction
a) Line 58-61: Please provide a brief rationale for the selection of the HBV genotype A source for the NASVAC vaccine.
b) Line 75: Replace "some some" with "some."
2.3 Materials and Methods
a) Lines 77-94: Please provide a brief description of the control group (PegIFN treated patients) in terms of dosage and administration route.
b) Lines 103-113: It would be helpful to provide a brief description of the inclusion and exclusion criteria or reference the relevant publication.
c) Line 114: Please clarify if the total number of patients is 160 or 360, as there seems to be a discrepancy between the abstract and the Materials and Methods section.
d) Line 120-122: Please provide a rationale for the change in the administration route of NASVAC between the first and second cycles.
e) Line 126: Clarify the duration of the second cycle of NASVAC treatment.
f) Line 127-128: Specify the primary and secondary endpoints of the phase III clinical trial.
There are several typographical errors throughout the manuscript that should be corrected. Examples include "Dwere" (line 171), "Bnagladesh" (line 171), and "PRISM 5" (line 163). Please proofread the manuscript carefully to ensure proper spelling and grammar.
In the Results section, the authors should specify the unit of measurement for the ALT levels mentioned throughout the text (e.g., U/L).
The authors should consider providing a more detailed explanation of the abbreviations used in the manuscript, such as EOT (end of treatment) and ULN (upper limit of normal), for the benefit of readers who may not be familiar with these terms.
Author Response
Comments and Suggestions for Authors
Title: Antiviral response across genotypes after treatment of chronic hepatitis B patients with the therapeutic vaccine NASVAC or Pegylated interferon
General Comments
This study aims to evaluate the antiviral effects of NASVAC, a therapeutic vaccine for chronic hepatitis B (CHB), compared to Pegylated interferon (PegIFN) in a phase III clinical trial. The study also provides information about the role of HBV genotype on the therapeutic outcomes. The paper is well-written, and the findings are valuable for the field of hepatitis B treatment. Furthermore, this manuscript presents the results of a clinical trial comparing the safety and efficacy of HeberNasvac, a therapeutic vaccine for chronic hepatitis B, with pegylated interferon (PegIFN) treatment. The authors evaluated the safety and efficacy of the treatments in patients with different genotypes of hepatitis B virus (HBV) and assessed their relationship with viral genotype, HBV DNA levels, and alanine aminotransferase (ALT) levels.
However, several aspects of the study should be addressed and clarified to improve the overall quality of the paper. Please find my specific comments and recommendations below.
Recommendations
The study is interesting and presents valuable findings; however, several points need to be addressed and clarified. The authors should:
Address the specific comments listed above.
Provide a rationale for the change in the administration route of NASVAC between the first and second cycles.
Clarify the duration of the second cycle of NASVAC treatment and provide the timepoints at which the HBV DNA reduction below 250 copies per mL was observed for NASVAC compared to Peg-IFN.
Specify the primary and secondary endpoints of the phase III clinical trial.
Response:
The overall objective of study design was to assess the safety and efficacy of NASVAC in CHB patients. The mode of therapy via nasal route was chosen due to several factors like targeting the mucosal immunity-based vaccine production. In addition, nasal route was selected as this can be used easily in developing and resource-constrained countries, as only injection-based vaccines cannot be used by the majority of population of those countries. But, these countries harbor the main bulk of CHB patients. Thus, treating these patients by non-parental mode is an important variable to attain the goal of treatment of CHB patients and attainment of “Elimination of Hepatitis by 2030”, a goal of WHO. The injection route was chosen to activate the systemic compartment of the patients by specific antigens. In fact, the safety and efficacy of both nasal and injection routes were optimized in a phase I/II clinical trial in which the mechanism of action was also studied in peripheral blood lymphocytes and antigen-presenting dendritic cells (Reference 9). This has been described in Method section (Line 109-120).
The viral load reduction was taken as its primary outcome. The success criterion was in relation with the proportion of patients showing reduction of the viral load under the limit of detection (250 copies/mL) after 24 weeks of each treatment completion, corresponding to weeks 48 and 72 for NASVAC and Peg-IFN, respectively. The secondary outcomes were: biochemical response as measured by the serum ALT transaminase level, evaluated every 12 weeks; serological response as measured by HBsAg detection and their specific antibodies (weeks 0, end of treatment and end of follow-up); serum HBeAg detection and its conversion to anti-HBeAg antibodies (weeks 0, end of treatment and end of follow-up); histological response as measured by Fibroscan (week 0 and week 96)
Major Comments:
2.1. The study design and methodology are generally well-described, but the authors should provide more information on the sample size calculation and the inclusion and exclusion criteria for the patient population.
Response: The present study is part of the efficacy trial of NASVAC, indexed in Clinicaltrials.gov and previously published, the most important criteria are mentioned in the text and also, we gave the reference to the index (NCT01374308) that describes all inclusion/exclusion criteria for patient selection. The previously published article that contains the details about the calculation of the sample size is indexed as reference 10 (Al Mahtab M, Akbar SMF, Aguilar JC, Guillen G, Penton E, Tuero A, Yoshida O, Hiasa Y, Onji M. Treatment of chronic hepatitis B naive patients with a therapeutic vaccine containing HBs and HBc antigens (a randomized, open and treatment controlled phase III clinical trial). PLoS One. 2018 Aug 22;13(8):e0201236. doi: 10.1371/journal.pone.0201236. eCollection 2018. Reference 10) and in this way the reader is informed. The study of efficacy across genotypes was not calculated to find significant differences; however, it was important to find statistical differences even when the study was not primarily designed for this aim. These have been shown in the revised manuscript (Line 143-176; Line Line 203-214; Line 219-221)
2.2. The authors should provide more information on the safety profile of the HeberNasvac vaccine, including the incidence of adverse events and any serious adverse events. A comparison of the safety profile between the vaccine and the PegIFN treatment group would be useful.
Response: More information safety profile has been provided. At the time of 24 weeks after end of treatment (EOT), the incidences of some adverse events were significantly higher in patients receiving Peg-IFN than those received NASVAC (Line 182-87; Line 236-239).
2.3. The authors should provide the specific p-values for the statistical analyses throughout the manuscript. Additionally, it would be helpful to provide exact numbers and percentages for the results presented in the text, particularly regarding the differences in the proportion of patients with each genotype in the treatment groups.
Response: Genotype D comparison: Fisher’s exact test p=0.0039; 16 out of 22 were HBV undetectable while 8 out of 28 were undetectable in the case of Peg-IFN Gen D patients. The comparison for other genotypes did not evidences significant differences (Line 265-268).
Minor Comments:
2.1 Abstract
- Line 17: Replace "sirface" with "surface."
Response; This has been done
- Line 21: Please clarify the timepoint at which the stronger antiviral effect (HBV DNA reduction below 250 copies per mL) was observed for NASVAC compared to PegIFN.
Response: 24 weeks after EOT.
- Line 25: Change "alteranative" to "alternative."
Response; This has been done
We are really sorry for the topographical errors. To address this, the final revise version of the manuscript was checked by MDPI English checking system.
2.2 Introduction
- Line 58-61: Please provide a brief rationale for the selection of the HBV genotype A source for the NASVAC vaccine.
Response: NASVAC has been prepared in Center for Genetic Engineering and Biotechnology (CIGB), Havana Cuba by mixing two antigens of HBV HBsAg and HBcAg. CIGB is endowed with a prolonged experience with development of vaccines against HBV. The vaccine developed from HBV genotype A has been proved to be safe and efficacious. Based on these experiences, HBV genotype A was selected to develop NASVAC.
- Line 75: Replace "some some" with "some."
Response: The entire manuscript has been checked by the English Checking Center as proposed by the Editors and Reviewers.
2.3 Materials and Methods
- Lines 77-94: Please provide a brief description of the control group (PegIFN treated patients) in terms of dosage and administration route.
Response; The dose of Peg-IFN was 180 µg, subcutaneously, once a week for 48 consecutive weeks.
- Lines 103-113: It would be helpful to provide a brief description of the inclusion and exclusion criteria or reference the relevant publication.
Response: A short description of inclusion and exclusion has been provide in the method section.
- Line 114: Please clarify if the total number of patients is 160 or 360, as there seems to be a discrepancy between the abstract and the Materials and Methods section.
Response; This has been done. The total number of patients were 160 in the phase III trial.
(d) Line 120-122: Please provide a rationale for the change in the administration route of
NASVAC between the first and second cycles.
Response: The overall objective of study design was to assess the safety and efficacy of NASVAC in CHB patients. The mode of therapy via nasal route was chosen due to several factors like targeting the mucosal immunity-based vaccine production. In addition, nasal route was selected as this can be used easily in developing and resource-constrained countries, as only injection-based vaccines cannot be used by the majority of population of those countries. But, these countries harbor the main bulk of CHB patients. Thus, treating these patients by non-parental mode is an important variable to attain the goal of treatment of CHB patients and attainment of “Elimination of Hepatitis by 2030”, a goal of WHO. The injection route was chosen to activate the systemic compartment of the patients by specific antigens. In fact, the safety and efficacy of both nasal and injection routes were optimized in a phase I/II clinical trial in which the mechanism of action was also studied in peripheral blood lymphocytes and antigen-presenting dendritic cells.
(e) Line 126: Clarify the duration of the second cycle of NASVAC treatment.
Response; In the second cycle, NASVAC was administered five times via both nasal route and subcutaneous route.
- Line 127-128: Specify the primary and secondary endpoints of the phase III clinical trial.
Response; The viral load reduction was taken as its primary outcome. The success criterion was in relation with the proportion of patients showing reduction of the viral load under the limit of detection (250 copies/mL) after 24 weeks of each treatment completion, corresponding to weeks 48 and 72 for NASVAC and Peg-IFN, respectively. The secondary outcomes were: biochemical response as measured by the serum ALT transaminase level, evaluated every 12 weeks; serological response as measured by HBsAg detection and their specific antibodies (weeks 0, end of treatment and end of follow-up); serum HBeAg detection and its conversion to anti-HBeAg antibodies (weeks 0, end of treatment and end of follow-up); histological response as measured by Fibroscan (week 0 and week 96).
There are several typographical errors throughout the manuscript that should be corrected. Examples include "Dwere" (line 171), "Bnagladesh" (line 171), and "PRISM 5" (line 163). Please proofread the manuscript carefully to ensure proper spelling and grammar.
Response; These have been corrected
In the Results section, the authors should specify the unit of measurement for the ALT levels mentioned throughout the text (e.g., U/L).
Response: This has been done.
The authors should consider providing a more detailed explanation of the abbreviations used in the manuscript, such as EOT (end of treatment) and ULN (upper limit of normal), for the benefit of readers who may not be familiar with these terms.
Response: This has been done and a list of abbreviation has been provided (Line 336-343).

Round 2
Reviewer 1 Report
Genotype names are inconsistently (Gen A, GA, Genotype-A, Genotype A) shown in the figures as well in the text.
The spelling of across is misspelt in the figure legend of Figure 2. Nasvac can be consistently shown as NASVAC in the Figure 2 legend.
Author Response
Genotype names are inconsistently (Gen A, GA, Genotype-A, Genotype A) shown in the figures as well in the text.
The spelling of across is misspelt in the figure legend of Figure 2. Nasvac can be consistently shown as NASVAC in the Figure 2 legend.
Response: Genotype names have been explained in Figure 1 and 2 in Figure legends (Lines 276-279 and Lines 289-291. The spelling has been corrected.

Reviewer 2 Report
The revised version is greatly improved.
Author Response
The revised version is greatly improved.
Response: Thank you for your kind understandings.

Reviewer 3 Report
Thanks to the authors for their careful revisions. This is now a much more carefully described study.
Author Response
Thanks to the authors for their careful revisions. This is now a much more carefully described study.
Response: Thank you very much for your appreciation.
